# Efficient generation of neutral and charged biexcitons in encapsulated WSe$_2$ monolayers

Ziliang Ye[1,2], Lutz Waldecker [1], Eric Yue Ma [1], Daniel Rhodes[3], Abhinandan Antony[3], Bumho Kim[3], Xiao-Xiao Zhang[1], Minda Deng[1], Yuxuan Jiang[4], Zhengguang Lu[4,5], Dmitry Smirnov [4], Kenji Watanabe [6], Takashi Taniguchi[6], James Hone[3] & Tony F. Heinz[1,7]

Higher-order correlated excitonic states arise from the mutual interactions of excitons, which generally requires a significant exciton density and therefore high excitation levels. Here, we report the emergence of two biexcitons species, one neutral and one charged, in monolayer tungsten diselenide under moderate continuous-wave excitation. The efficient formation of biexcitons is facilitated by the long lifetime of the dark exciton state associated with a spin-forbidden transition, as well as improved sample quality from encapsulation between hexagonal boron nitride layers. From studies of the polarization and magnetic field dependence of the neutral biexciton, we conclude that this species is composed of a bright and a dark excitons residing in opposite valleys in momentum space. Our observations demonstrate that the distinctive features associated with biexciton states can be accessed at low light intensities and excitation densities.

[1] Department of Applied Physics, Stanford University, 348 Via Pueblo Mall, Stanford, CA 94305, USA. [2] Department of Physics and Astronomy, University of British Columbia, Vancouver, BC V6T 1Z4, Canada. [3] Department of Mechanical Engineering, Columbia University, New York, NY 10027, USA. [4] National High Magnetic Field Laboratory, Tallahassee, FL 32310, USA. [5] Department of Physics, Florida State University, Tallahassee, FL 32306, USA. [6] National Institute for Materials Science, 1-1 Namiki, Tsukuba 305-0044, Japan. [7] SLAC National Accelerator Laboratory, Menlo Park, CA 94025, USA. Correspondence and requests for materials should be addressed to T.F.H. (email: tony.heinz@stanford.edu)

The strong Coulomb interaction in two-dimensional transition metal dichalcogenide (TMDCs) crystals, arising from reduced dielectric screening in the atomically thin limit, significantly modifies the optical properties of these materials and gives rise to pronounced excitonic effects[1]. The Coulomb attraction between electrons and holes in these systems leads to the formation of tightly bound excitons, which causes a reduction in the optical band gap by a large fraction of an electron volt and yields very short intrinsic radiative lifetimes[2–5]. Both trion (charged exciton) and biexciton (exciton molecule) states have been observed in TMDCs[6–8]. Here we report on the characterization of a second biexciton species in monolayer WSe$_2$, in addition to the previously observed species[7], which we identify as the neutral and negatively charged biexciton (four-particle and five-particle) states, respectively. In our high-quality samples produced using encapsulation in hexagonal boron nitride (hBN), local inhomogeneities and nonradiative decay channels in the monolayer semiconductor are strongly reduced. As a consequence, the emission of the neutral and charged biexciton is observed at an excitation density three orders of magnitude lower than in previous reports. By comparing the emission of the different exciton species under strong magnetic fields and as a function of doping level, we deduce that the neutral biexciton is formed predominantly from the combination of a dark (spin-forbidden) and bright (spin-allowed) exciton residing in opposite valleys in momentum space. Our observations provide new understanding of many-body excited states in 2D semiconductors and open a path for utilizing the features of biexciton states for quantum and optoelectronic applications at low light intensities.

The lowest lying interband transition in monolayer tungsten compounds of the TMDC family has been identified as spin-forbidden[9]. The exciton associated with the lower conduction band therefore, to first order, does not emit light; it is referred to as a dark exciton. The existence of dark excitons strongly influences the optical properties of the material, including the photoluminescence (PL) quantum efficiency at low temperatures[10], the ultranarrow linewidth of the PL emission along the in-plane direction[11], the efficient coupling to surface plasmon polaritons[12], and the large tip-induced enhancement of the PL[13]. Because the dark exciton has a lifetime that is orders of magnitude longer than the bright exciton at low temperature[14], the steady-state density of dark excitons can be much higher than that of bright excitons for continuous-wave excitation. Correspondingly, the diffusion length of the dark exciton is also enhanced. When a dark exciton approaches another exciton, the two excitons can interact to form a biexciton (Fig. 1a). Therefore, the formation of biexcitons via the long-lived dark exciton states is much more likely than through two short-lived bright excitons. Here we report the observation of a neutral and charged biexciton composed of both bright and dark excitons.

## Results

**Observation of biexcitons in the charge-neutral regime**. We observe the evidence of a biexciton in our high-quality encapsulated WSe$_2$ samples (see Methods). In these samples, the defect density of the TMDC is reduced by about two orders of magnitude ($5 \times 10^{10}$ cm$^{-2}$) by exfoliating flux-grown crystals. The observed bright exciton linewidth (FWHM 4.5 meV at 20 K) is approaching the radiative limit[15]. In the low-temperature PL spectrum, the spin-allowed bright exciton emission appears at 1.723 eV (Fig. 1c). Emission from the spin-forbidden dark exciton with a linewidth of only 0.8 meV and an out-of-plane dipole radiation pattern (Supplementary Fig. 1) is observed at 1.681 eV, in agreement with previous reports[11,12]. As the excitation power is increased, a new peak emerges at

1.703 eV, lying 20 meV below the bright exciton peak. Compared to the emission of the bright and dark exciton species, which rise nearly linearly with laser intensity, the new peak increases approximately quadratically. A power-law fit of the emission yields an exponent of $1.96 \pm 0.03$ (Fig. 1d). The quadratic intensity dependence suggests that the peak arises from a few-body state formed from two excitons, each exhibiting a concentration growing linearly with intensity (Supplementary Note 2). The new biexciton peak, we should note, is observed for rather weak continuous wave (cw) excitation, with an irradiance of $1.4 \times 10^3$ W/cm$^2$, more than three orders of magnitude lower than in typical experiments employing pulsed laser excitation[7]. The asymmetric peak X$^i$, lying 32 meV below X$^0$, and the series of peaks around 1.665 eV have been attributed to indirect (momentum-dark) excitons[16]. The lower-lying emission peaks also originate from the WSe$_2$ sample, but have an unknown origin and are beyond the scope of this work.

**Spin and valley configuration of the biexciton**. In the following, we focus on identifying the composition of the newly observed biexciton state. Regarding the spin degree of freedom, biexcitons can be classified as composed of dark–dark, dark–bright, and bright–bright exciton components. (Here we assume the electron and hole of the constituent excitons remain in the same band when forming a biexciton.) The emission energy of the biexciton precludes it from being composed of two dark excitons, as this would require a negative binding energy. The bright–bright configuration is unlikely to occur given our experimental conditions: At the greatest applied laser intensity, we estimate an upper limit of 52 µm$^{-2}$ of bright excitons in the steady state, for a 2-ps lifetime of the bright exciton[5] and assuming a 100% conversion efficiency from absorbed photons to bright excitons. This density is smaller than the dark exciton density at our lowest excitation level, as the dark exciton has a lifetime more than 50 times longer than the bright species, as reported by Robert et al.[14] and confirmed in our sample. The low density of bright excitons renders the bright–bright biexciton much less likely to form. (A simple kinetic model is presented in Supplementary Note 3 to describe the biexciton formation process.) We therefore conclude that the observed biexciton is a combination of one bright and one dark exciton (Fig. 1b). This assignment is further supported by the magnetic field and doping dependences of the biexciton emission, as discussed below.

To investigate the biexciton valley configuration further, we apply a strong out-of-plane magnetic field, which acts to break the valley degeneracy. As a magnetic field along the normal direction of the sample shifts the conduction and valence bands in the different valleys in opposite directions, the exciton peaks are split into pairs[17–19] associated with their valley index. Since the dark exciton has a total spin angular momentum of 1, its Zeeman splitting is larger than that of the bright exciton[14] with its vanishing total spin. As the emission of the biexciton results from its bright component, leaving behind a dark exciton in the final state, the biexciton emission energy is given by the bright exciton emission energy reduced by the biexciton binding energy $\Delta_{XX}$ (see Methods). Neglecting any difference in magnetic field dependence of the (small) biexciton binding energy $\Delta_{XX}$, we predict within our picture that the XX peak should exhibit the same g-factor as the bright exciton. This agrees with our experimental observations: X$^0$ and XX have g-factors of $4.4 \pm 0.5$ and $4.6 \pm 0.5$, respectively, while X$^D$ has a g-factor of $9.6 \pm 0.5$ (Fig. 2a).

The distribution of the exciton population between the different branches of the Zeeman-split pairs will, in thermal equilibrium, follow a Boltzmann distribution. This suggests that

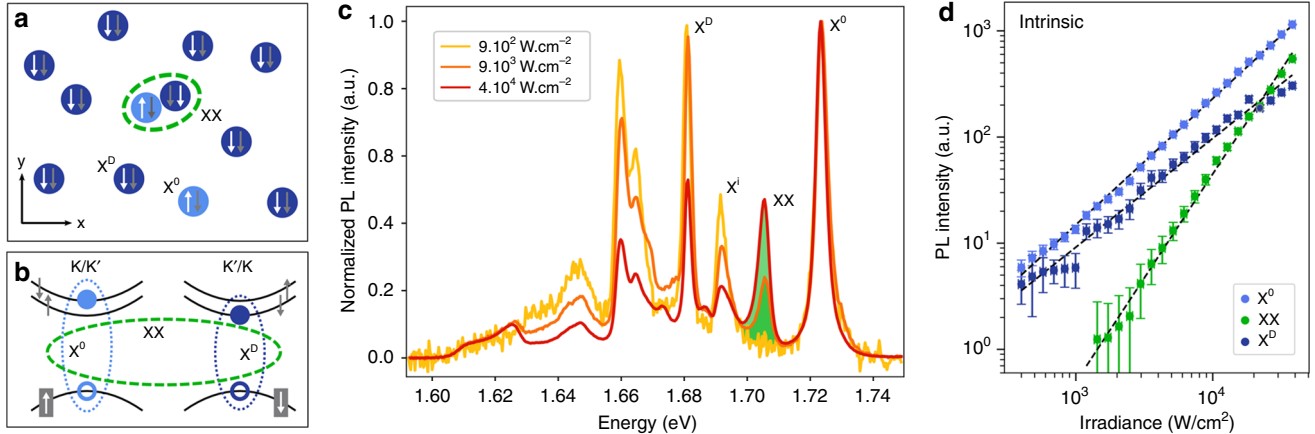

**Fig. 1** Observation of a biexciton composed of a bright (spin-allowed) and a dark (spin-forbidden) exciton. **a** Schematic of biexciton formation in real space. Because the dark exciton ($X^D$, dark blue) has a total spin of 1, it has a much longer radiative lifetime and therefore a much higher density than the bright exciton ($X^0$, light blue), which has a total spin of zero. A biexciton (XX, green circle) can form as a bright and a dark exciton approach one another. **b** Representation of the biexciton in momentum space. **c** Photoluminescence (PL) spectra of an encapsulated $WSe_2$ monolayer sample for different excitation intensities. All spectra are normalized to the $X^0$ peak. A new peak (green shading) emerges at high excitation levels and is attributed to biexciton emission. The XX peak is located at 1.703 eV, 20 meV below the $X^0$ peak. The $X^D$ peak at 1.681 eV is identified by examining its distinct radiation pattern. Other peaks are discussed in the main text. **d** Power dependence of the emission of different excitonic species. Datapoints and errorbars are results of pseudoVoigt fits to the raw spectra. The XX peak grows superlinearly with incident intensity with an exponent of 1.96. In contrast, the $X^0$ and $X^D$ peaks have exponents of 1.22 and 1.03, respectively

even for a partially thermalized system, which starts from equal populations when degenerate, the low-energy branch (LEB) of the pair should have a larger population, and thus emit more light, than the high-energy branch (HEB). This relation is satisfied for the pairs of dark excitons and most of the features seen in the PL. However, for the bright exciton pair, the emission intensity is comparable, and in some conditions inverted. Such an unusual distribution might arise from the efficient intervalley scattering between two conduction bands of same spin (see Supplementary Note 4 for a detailed discussion). More importantly, the emission intensity from the biexciton pair is also observed to be inverted, with an inversion even more pronounced than for the bright exciton. This observation suggests the biexciton to be composed of a bright and a dark exciton residing in opposite valleys. In the intervalley configuration (Fig. 2c), the dark component in the HEB (LEB) of the biexciton is associated with the LEB (HEB) of the dark exciton[17]. We expect that the rate of biexciton formation should scale as the product of the constituent exciton densities, that is $\propto \rho_{X^D,L(H)} \cdot \rho_{X^0,H(L)}$. If the density of biexcitons $\rho_{XX,H(L)}$ is much smaller than that of bright excitons, after being normalized to their lifetimes, the biexciton density will scale in the same fashion (see Supplementary Note 2). With the experimentally observed larger population of the dark exciton in the LEB, an intervalley configuration of the biexciton leads naturally to an inverted distribution for the biexciton emission. More quantitatively, the LEB/HEB ratio of the biexciton pair should be close to the quotient of the ratios observed in the bright and dark exciton pairs, a result consistent with our experimental findings at different magnetic fields (Fig. 2b).

The fact that the intervalley configuration of the biexciton is favored over other configurations can be understood in terms of energetics. The intravalley configuration—bright and dark excitons residing in the same valley—needs to involve two holes sharing the same valence band, which costs extra energy as Pauli blocking forces the second hole to occupy a higher momentum/energy state (the footprint of an exciton in

momentum space can be approximated by the inverse of the exciton Bohr radius of about 1 nm). Another possible configuration would be that of a biexciton composed of a direct and an indirect exciton—two electrons are located in different bands of the same valley, but with the two holes in opposite valleys. This configuration costs extra exchange energy, which should be about 10 meV as inferred from the peak separation between the two negatively charged trions[20,21]. Within our experiment, we cannot, however, exclude potential contributions from Q point indirect excitons[22].

**Distinguishing neutral and negatively charged biexcitons.** Finally, we are able to elucidate the relationship between the newly observed and the previously reported biexciton[7] by tuning the static carrier density in the material (Fig. 3a). In the low excitation regime, the emission spectra agree well with those reported in the literature: the bright and dark excitons are observed in the intrinsic regime while different types of trions emerge in the n-doped and p-doped regimes[20,21]. Unlike the bright exciton state, which extends to finite doping levels with an asymmetric gate dependence, the dark exciton is strictly limited to the intrinsic regime. Under stronger excitation conditions ($\sim 4 \times 10^3$ W cm$^{-2}$), the biexciton peak at 1.703 eV appears, and its emission is also restricted to the intrinsic regime. The doping dependence therefore confirms that this newly observed XX peak results from a neutral biexciton formed by a bright and a dark exciton, as it appears only in the doping regime where both types of excitons coexist (Fig. 3b).

Under strong excitation conditions, the emission of another biexciton is observed in our spectra, which exhibits a superlinear power dependence with an emission strength scaling with the pump laser intensity as $I^{1.56}$ (Supplementary Fig. 2). This peak lies 52 meV below the bright exciton and has been reported in the literature[7]. However, this biexciton only appears at the crossover from the intrinsic to the n-doped regime (Fig. 3b), suggesting that it is a negatively charged biexciton XX$^-$, formed from a trion and a neutral exciton, as opposed to its earlier assignment as a neutral biexciton[7]. In contrast to the negatively charged trions, no fine

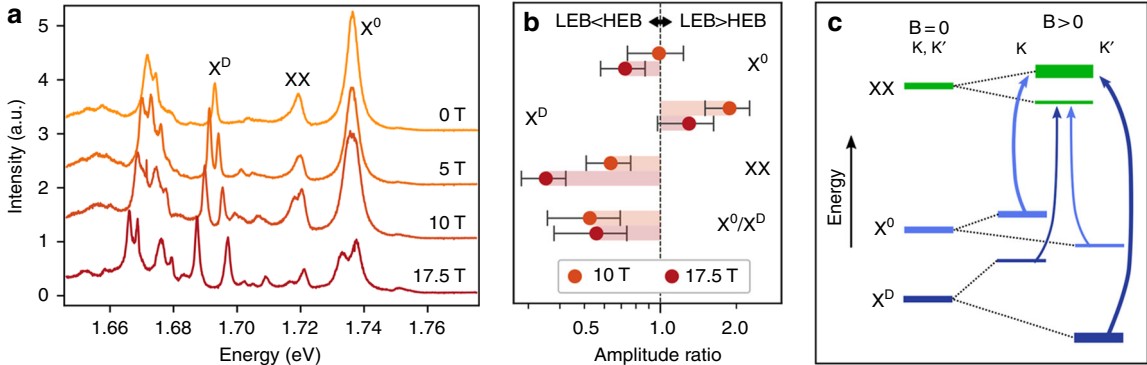

**Fig. 2** Zeeman splitting and non-equilibrium distribution of biexciton emission. **a** A strong out-of-plane magnetic field splits most emission peaks into pairs. The splitting of XX is similar to that of the $X^0$ feature (g-factor 4.4 ± 0.5), whereas the splitting of the $X^D$ feature is larger because of its non-zero spin. **b** The ratios of amplitudes of the fitted low-energy branch (LEB) and high-energy branch (HEB) of the emission spectra. Unlike the $X^D$ pair, the XX emission pair exhibits inverted populations between the HEB and LEB. The LEB/HEB ratio of the XX pair is similar to that of $X^0/X^D$. The error bars for $X^0$, $X^D$, and XX are conservative estimates. **c** A diagram of the biexciton intervalley configuration as probed through the PL measurements in a magnetic field. The B-field splits the bright (light blue) and dark (dark blue) exciton into pairs according to their valley index ($K$, $K'$). Analysis of the PL intensities shows that the biexciton is composed of excitons from different valleys

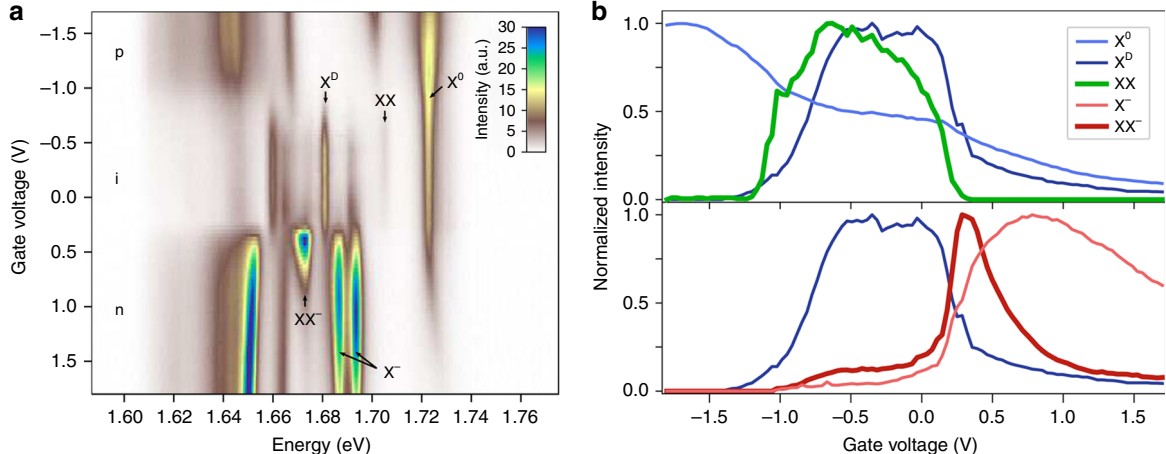

**Fig. 3** Doping dependence of the PL emission of neutral and charged biexcitons. **a** Gate dependence map of the PL intensity above the onset for observing both biexciton species ($I_0 = 4 \times 10^3$ W cm$^{-2}$). The intrinsic regime is characterized by the strong emission from $X^D$. In the n-doped and p-doped regimes, trions and other species emerge. The XX peak at 1.703 eV is only observed in the intrinsic regime, indicating a neutral biexciton composed of a neutral $X^0$ and $X^D$. Another peak with superlinear power dependence is observed at 1.671 eV, 52 meV below $X^0$. It is assigned to a charged biexciton state, as it appears only in the low n-doped regime. **b** Normalized PL intensity of the exciton species relevant for the formation of biexcitons as a function of doping level. The neutral biexciton (green) emission is limited to the intrinsic regime, in which $X^D$ (dark blue) is observed, showing the important role of $X^D$ in the formation of XX. The $XX^-$ emission (dark red) is strongest in the low n-doped regime, where $X^0$ (light blue), $X^D$ and the two $X^-$ species coexist (light red, only lower $X^-$ shown for clarity)

features can be resolved in the $XX^-$ peak at this stage. If no more than one electron/hole resides in any band, there should be two spin-valley configurations for $XX^-$. Due to the conduction band splitting, we expect the energetically favored configuration to be two holes in opposite valence band valleys, two electrons in opposite lower conduction band valleys and one electron in the higher conduction band. Because the valence band splitting is much larger —about 400 meV—no positively charged biexciton ($XX^+$) emission would be observed. Since the $XX^-$ peak appears most prominently at relatively low doping levels, we can rule out the possibility of it being related to a plasmon coupled exciton[23]. With respect to the two negatively charged trions, the $XX^-$ has a binding energy of 14 or 21 meV, in agreement with theoretical calculations[24].

## Discussion

The identification of neutral and charged biexcitons resolves the discrepancy of the biexciton binding energy found between previous experiments and theories: as calculated, the biexciton binding energy is indeed smaller than the trion binding energy due to the form of the screened Coulomb potential in two-dimensional TMDCs[24–28]. The 20-meV binding energy also matches that of the neutral biexciton in MoSe$_2$ monolayers observed by Hao et al. using two-dimensional coherent spectroscopy[8] and of WSe$_2$ in pump-probe spectroscopy[29], even though both works describe bright–bright excitons, which would suggest the biexciton binding energy is only weakly sensitive to the spin configuration of the two constituent excitons.

In conclusion, we have observed neutral and charged biexciton species in monolayer $WSe_2$ crystals encapsulated in hBN through their clear emission signatures. We have established that the neutral biexciton is composed of pairs of excitons located in opposite valleys, one of which is a long-lived dark species. Our observations deepen the understanding of the intrinsic optical properties of two-dimensional TMDCs and provides a new route to access the optical response associated with biexciton species under conditions of modest excitation.

## Methods

**Samples**. Single crystals were grown by sealing W powder, 99.999% purity, and Se shot, 99.999% purity, in an evacuated quartz ampoule with excess Se as the flux. Subsequently, the ampoule was heated to 1000 °C for 2 days and cooled to 450 °C at a rate of 0.85 °C h$^{-1}$. The crystals were removed, centrifuged, and annealed at 250 °C for 2 days to remove excess Se. Encapsulated monolayers of $WSe_2$ were then prepared from these single crystals by a dry stamping technique[30], with all materials exfoliated on $Si/SiO_2$ substrates. In order to contact the $WSe_2$ film, two few-layer graphene strips were picked up by an hBN flake, followed by the monolayer $WSe_2$. Finally, a bottom ($Ar/O_2$ cleaned) hBN crystal and a thin layer of graphite were picked up to act as the backgate for controlling doping and screening the effects of the $SiO_2$. Contact was made to the graphite leads and backgate by etching with $CHF_3$ and making edge contact using Cr/Pd/Au film deposition (2/25/50 nm).

**Measurements**. In all but the magnetic field experiments, the samples were mounted in a continuous flow optical cryostat and cooled with liquid helium to temperatures of ~15 K. PL spectra were taken with an imaging spectrometer after 640 nm cw laser excitation. We note that for laser excitation at shorter wavelengths (e.g., 532 nm), we observed photodoping of the sample starting at moderate power levels. Spectra on bare hBN were taken to verify that all features in Fig. 1b originate from the $WSe_2$ sample. Magnetic field measurements were conducted at the National High Magnetic Field Laboratory, at sample temperatures of 10 K and cw laser excitation at 660 nm. The PL spectra were fitted with a series of psedudoVoigt lineshapes to obtain the emission intensity (area) as a function of the relevant experimental parameters.

**Biexciton binding energy**. The binding energy $\Delta$ of the neutral biexciton is obtained from the relation for energy conservation in the formation and radiative decay of the biexciton species: $E_{X^0} + E_{X^D} - \Delta = \hbar\omega_{XX} + E_{X^D}$. Here we consider a biexciton formed from a pair of bright and dark excitons and assume that the final state after radiative decay is a dark exciton in its ground state. It follows that the biexciton binding energy is directly related to experimentally observable emission energies through $\Delta = E_{X^0} - \hbar\omega_{XX}$.

## Data availability

The data supporting the plots within this paper and other findings of this study are available from the corresponding author upon request.

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

## Acknowledgements

The authors thank Timothy Berkelbach and Ting Cao for fruitful discussions. The spectroscopic studies were supported by the Department of Energy, Office of Science, Basic Energy Sciences, Materials Sciences and Engineering Division, under Contract DE-AC02-76SF00515 for the magnetic field measurements and by the Gordon and Betty Moore Foundation's EPiQS Initiative through Grant no. GBMF4545 for analysis. Sample preparation was supported by the National Science Foundation MRSEC program through Columbia in the Center for Precision Assembly of Superstratic and Superatomic Solids (DMR-1420634). L.W. acknowledges support by the Alexander von Humboldt Foundation. Y.J., Z.L. and D.S. acknowledge support from the US Department of Energy (DE-FG02-07ER46451) for magneto-photoluminescence measurements performed at the National High Magnetic Field Laboratory, which is supported by the NSF Cooperative Agreement no. DMR-1157490 and the State of Florida.

## Author contributions

Z.Y., L.W. and T.F.H. conceived the project. D.R., A.A. and B.K. grew the $WSe_2$ crystals and prepared encapsulated and backgated samples. K.W. and T.T. produced the hBN crystals. Z.Y., L.W., E.Y.M., Y.J., X.Z. and M.D. conducted the measurements. Z.Y. and

L.W. analyzed the data. Z.Y., L.W. and T.F.H. wrote the manuscript with input from all authors.

## Additional information

**Competing interests:** The authors declare no competing interests.

