## [Peer Review File · Nature Communications]

Reviewers' comments:

Reviewer #1 (Remarks to the Author):

In the manuscript "Efficient generation of neutral and charged biexcitons in encapsulated WSe₂ monolayers", Z. Ye and coauthors report the observation of photoemission resulting from neutral and charged biexcitons in encapsulated monolayers of WSe₂. Biexcitons in TMDs are an intensely discussed topic, in a particular as theoretical results and experiments seemed to disagree about the binding energies. The content of the manuscript is thus timely and interesting for a broad audience. Before I can recommend publication in Nature Communications, however, I have several questions I ask the authors to address:

- (1) The sample is a monolayer of WSe₂ encapsulated in BN. BN is known to show rich spectroscopic features itself and to be prone to have defects. Can the authors provide evidence that the states they observe are intrinsic to WSe₂ and do not originate from the contact with BN encapsulation?
- (2) Also along these lines, polarization resolved spectra are usually used to conclusively identify biexciton complexes. The manuscript would profit much from the inclusion of such data.
- (3) The quadratic scaling of luminescence shown in Fig. 1d is not immediately obvious for a biexciton composed of two different entities with possibly intensity dependencies (see Fig. 1d) and lifetimes. Please provide a derivation of the expected scaling with intensity including both dark and bright exciton populations.
- (4) The authors observe luminescence also from the dark exciton with their high NA objective. With the exciton population presumably concentrated in the dark state, why there is no signature of a dark-dark biexciton? Could the feature at 1.66 eV in Fig. 3 be possibly such a signature?
- (5) Two of the authors have also been involved in the study published in Ref. 7 of the manuscript. There, the complex now related to XX⁻ was identified as four-particle complex (neutral biexciton), and this assignment has been supported by theoretical calculations. Reference 7 is cited many times throughout the manuscript, but the previous assignment as a neutral exciton is not mentioned. To avoid misunderstandings, could the authors clarify this point and briefly discuss possible shortcomings of the theory employed in Ref. 7?
- (6) Based on data obtained from non-encapsulated samples, many authors have speculated about the XX binding energy being less than the trion binding energy, e.g. Nano Lett. 2014, 14, 202–206; Physical Review B 92, 125417, 2015, Nano Lett. 16, 2945, 2016, Nat. Commun. 8, 1, 2017, arxiv:1801.04225. In this context, I ask the authors not to state that they "discovered" this particular state.

Reviewer #2 (Remarks to the Author):

The manuscript by Ye and co-workers reports on the observation of neutral and charged biexcitons in monolayer WSe₂ encapsulated in hBN. This work is timely and interesting, as many-particle complexes in TMDs are currently intensively discussed in the literature. In particular, the first identification of the biexciton in this system was controversial as the experimentally determined binding energy was much larger than the theoretically predicted value. Therefore, a clear-cut identification of this spectral feature is highly desirable and the results would be very interesting to scientists within the field.

I have some questions and remarks, which I ask the authors to address before I recommend a publication in Nature Communications:

- (1) A crucial point is to ensure that the luminescence measured between exciton and trion is an intrinsic property of the material. Can the authors give evidence that this emission is not related to defects or a result of the specific choice of the environment? In particular, hBN is known to obey a rich defect-related spectrum itself.

(2) For the biexcitons, similar observations have been made recently for monolayer WS₂ (Nagler arXiv:1801.09255) as well as in pump probe experiments on WSe₂ (Steinhoff et al. arXiv:1801.04225). I think these two works should be mentioned and deserve a discussion, especially as their interpretation is different.

(3) The resonance the authors identified as a charged biexciton has recently been explained by the coupling of excitons to inter-valley plasmons (Tuan et al. PRX 7, 041040 (2017)). Can the authors rule out this alternative interpretation?

(4) Why does the X₀ transition broaden in case of p-doping and vanishes under n-doping?

(5) The central argument for the dark exciton contribution to the biexciton is that under magnetic field the higher Zeeman splitted transition is stronger populated. The authors also claim that the charged biexciton is formed by a trion and a bright exciton. According to their argument, I would expect a stronger low energy branch - is this indeed the case for the charged biexciton?

(6) In my opinion, it makes no sense to use biexcitons in TMDs as sources for entangled photons if the biexciton state is extended and not localized like in a quantum dot. Therefore, I find the statement related to Ref. 9 in the introduction a bit misplaced.

Reviewer #3 (Remarks to the Author):

The manuscript by Ye and co-workers reports on the observation of neutral and charged biexcitons in monolayer WSe₂ encapsulated into high quality hBN layers. The interpretation is supported by continuous wave photoluminescence experiments performed at low temperature as a function of excitation power, out of plane magnetic field and electrostatic doping. This work is timely as similar observations have been made in WS₂ monolayers by the group of Tobias Korn (arXiv 1801.09255) and the broad community of researchers working on TMDC will certainly find interest in this work.

I really appreciate the manuscript and how the authors provide additional explanations that raise open questions instead of leaving them under the carpet. Thus, I think these results will still be debated in the community as they seem to be in contradiction with the interpretation of Korn's group (arXiv 1801.09255) in WS₂ monolayer and with the pump probe experiments performed by Elaine Li's group (arXiv 1801.04225). I think that these two studies deserve to be mentioned in the text to nuance affirmative conclusions. Moreover, I have a few questions and suggestions before recommending publication in Nature Communications journal.

- The authors mention that the bright exciton also exhibits inverted population and discuss some possible mechanism in the supplement information. I was wondering if the authors did observe the same inverted polarization for the trion features.

- I think that Fig2b and 2c are not clear. For Fig2c, I suggest to put the biexciton at the good energy which is EX₀+EXD-E_{binding}. It will help the reader to understand why the amplitude ratio of XX is compared with the one of X₀/X_D on Fig2b.

- In the supplementary information, the authors calculate a rough estimate of the exciton density needed for biexciton formation. This is very interesting and I understand that this is just a rough estimation, but I think that the values of the diffusion coefficient extracted from the linewidth are highly questionable. I suggest to use the very recent values measured in this publication (arXiv 1802.09201 (2018)).

- Can the authors propose an explanation of the superlinearity of the trions (exponents 1.2 in the SI).

- Minor : in page 4, can the authors add the value of the measured FWHM ?

We thank all referees for their encouraging and constructive comments. In the following, we reply to their questions in a point-to-point fashion. The change in the manuscript has been highlighted accordingly.

Reviewer #1 (Remarks to the Author):

In the manuscript "Efficient generation of neutral and charged biexcitons in encapsulated WSe₂ monolayers", Z. Ye and coauthors report the observation of photoemission resulting from neutral and charged biexcitons in encapsulated monolayers of WSe₂. Biexcitons in TMDs are an intensely discussed topic, in a particular as theoretical results and experiments seemed to disagree about the binding energies. The content of the manuscript is thus timely and interesting for a broad audience. Before I can recommend publication in Nature Communications, however, I have several question I ask the authors to address:

(1) The sample is a monolayer of WSe₂ encapsulated in BN. BN is known to show rich spectroscopic features itself and to be prone to have defects. Can the authors provide evidence that the states they observe are intrinsic to WSe₂ and do not originate from the contact with BN encapsulation?

We agree with referees 1 and 2 that hBN shows rich photoluminescence features, many of which have been attributed to single photon emitters, and that defects in BN can influence the emission from WSe₂.

To address this concern, we have taken reference PL spectra in regions where no WSe₂ is present (only the encapsulating BN), but did not observe any emission feature within 10^{-3} of the PL intensity from WSe₂. All spectral features reported in the manuscript have been repeatedly observed at multiple points in multiple samples. As a further confirmation, the same observation is being reported in a parallel arXiv paper (arXiv 1802.10247). We have added a few words in the main text and a sentence in the Methods section to clarify this aspect.

On the other hand, one of the reasons for conducting PL measurements with an excitation wavelength of 640 nm is that we indeed observe a very strong photodoping effect when exciting the samples with shorter wavelengths such as 532 nm, which suggests photo-induced ionization of defects in BN. We have also added a sentence in the methods about this photodoping effect.

(2) Also along these lines, polarization resolved spectra are usually used to conclusively identify biexciton complexes. The manuscript would profit much from the inclusion of such data.

Following the referee's suggestion, we have carried out a series of polarization resolved PL measurements. The measurements are done without magnetic field since we don't have magnet time in the national high magnetic field lab at this point, where previously the magneto PL data was taken. Clearly the XX and XX- emission in the figure below (panel a and b) have the same circular polarization as the excitation, which suggests their intrinsic nature and supports the main conclusion of our paper. In addition, we observe the circular polarization to appear over the whole PL spectrum of the intrinsically doped sample (panel a), except for the dark exciton emission, which cannot bear circular polarization for its out-of-plane dipole. This widely spread polarization is enhanced in the n doped regime and

suppressed in the p doped regime (panel b and c).

No linear polarization is observed in either the XX or XX-

peak.

We have added the above discussion and figures into the supplementary information.

3) The quadratic scaling of luminescence shown in Fig. 1d is not immediately obvious for a biexciton composed of two different entities with possibly intensity dependencies (see Fig. 1d) and lifetimes. Please provide a derivation of the expected scaling with intensity including both dark and bright exciton populations.

We thank the referee for raising this important question. Below is our derivation for the expected dependence of the biexciton density, and accordingly its emission intensity, over the excitation power. We have added this part of discussion to the supplementary material and mentioned in the main text.

We start from a series of rate equations determining the density of bright and dark exciton as well as the biexciton.

$$\begin{aligned}\frac{dn_1}{dt} &= -\frac{n_1}{\tau_1} - \gamma n_1 n_2 + P_1 \\ \frac{dn_2}{dt} &= -\frac{n_2}{\tau_2} - \gamma n_1 n_2 + \frac{n_3}{\tau_3} + P_2 \\ \frac{dn_3}{dt} &= -\frac{n_3}{\tau_3} + \gamma n_1 n_2\end{aligned}$$

Here n_1, n_2, n_3 are the densities of X^0, X^D, XX , and τ_1, τ_2, τ_3 are their lifetimes. γ is the formation rate of XX , a constant to the first order approximation. P_1 and P_2 are the bright and dark exciton formation rates, which, according to our measurement in the low density regime, are proportional to the α exponent of incident power (I^α), where α is 1.22 and 1.03 for X^0 and X^D . We can solve the steady-state density of each exciton species as follows:

$$n_1 = \frac{P_1}{\gamma P_2 \tau_2 + \frac{1}{\tau_1}}, \quad n_2 = P_2 \tau_2, \quad n_3 = \frac{\gamma \tau_2 \tau_3 P_1 P_2}{\gamma P_2 \tau_2 + \frac{1}{\tau_1}}$$

The biexciton density is proportional to the product of the bright and dark exciton density, and is about quadratically dependent on the incident power if $\gamma P_2 \tau_2$ is much smaller than $\frac{1}{\tau_1}$, or equivalently $\frac{n_3}{\tau_3} \ll \frac{n_1}{\tau_1}$. Otherwise, the bright exciton density will saturate and the biexciton density will become linearly dependent on the power. Our experimental condition is clearly far away from the saturation regime as the bright exciton emission grows linearly with power and the biexciton emission is significantly weaker than the bright exciton even at the highest excitation intensity.

(4) The authors observe luminescence also from the dark exciton with their high NA objective. With the exciton population presumably concentrated in the dark state, why there is no signature of a dark-dark biexciton? Could the feature at 1.66 eV in Fig. 3 be possibly such a signature?

We agree with the referee that the dark-dark biexciton should exist in our sample due to the large density of dark excitons. At the moment, we cannot conclude on the dark-dark biexciton energy since no emission was observed with superlinear power dependence below the dark exciton peak. This is likely caused by the small oscillator strength (slow emission rate) of the spin-forbidden transition. The 1.66 eV peak is likely not a biexciton peak, mostly because of its linear to sublinear power dependence. Also, no out-of-plane dipole emission is observed in the 1.66-eV peak (Fig. 1 in SI).

(5) Two of the authors have also been involved in the study published in Ref. 7 of the manuscript. There, the complex now related to XX- was identified as four-particle complex (neutral biexciton), and this assignment has been supported by theoretical calculations. Reference 7 is cited many times throughout the manuscript, but the previous assignment as a neutral exciton is not mentioned. To avoid misunderstandings, could the authors clarify this point and briefly discuss possible shortcomings of the theory employed in Ref. 7?

The first shortcoming of the earlier paper was on the experimental side: the quality of the samples that were available in 2014, when the previous study was published, was significantly lower than those available to us now. This means the features had a much larger linewidth and most likely more defects, so that the neutral biexciton could not be observed.

On the theory side, a variational analysis was performed, which yielded a biexciton binding energy of 37 meV, and therefore lower than the (incorrect) experimentally obtained value of 52 meV. The authors speculated that the variational analysis might underestimate the binding energy by 40-50% because of an insufficient optimized trial wavefunction, which may have not been the case.

Secondly, the reason for the theory estimate of the biexciton binding energy having been 37 meV, and thus still too high, might be found in the screened Coulomb potential. More specifically, the value of the screening length r_0 in the Keldysh potential, which is sensitive to the dielectric screening within the 2D material as well as from the surrounding substrate (Chernikov et al., PRL 113, 076802), is critical for the prediction of the biexciton binding energy. The details of the screening were not, and are still not, completely understood. Therefore, there is a relatively large uncertainty related to the choice of r_0 , which was probably too low in the earlier calculation.

We mention the new assignment in the introduction and devote an entire paragraph on page 7/8 to the discussion on the difference between the two biexciton states. In the new version of the manuscript, we explicitly point out the incorrect assignment in our previous paper: "...suggesting it to be a negatively charged biexciton XX^- , formed by a trion and a neutral exciton, as opposed to the previous assignment as a neutral biexciton [7]".

(6) Based on data obtained from non-encapsulated samples, many authors have speculated about the XX binding energy being less than the trion binding energy, e.g. Nano Lett. 2014, 14, 202–206; Physical Review B 92, 125417, 2015, Nano Lett. 16, 2945, 2016, Nat. Commun. 8, 1, 2017, arxiv:1801.04225. In this context, I ask the authors not to state that they "discovered" this particular state.

We have removed the word "discovered" from the manuscript.

Reviewer #2 (Remarks to the Author):

The manuscript by Ye and co-workers reports on the observation of neutral and charged biexcitons in monolayer WSe₂ encapsulated in hBN. This work is timely and interesting, as many-particle complexes in TMDs are currently intensively discussed in the literature. In particular, the first identification of the biexciton in this system was controversial as the experimentally determined binding energy was much larger than the theoretically predicted value. Therefore, a clear-cut identification of this spectral feature is highly desirable and the results would be very interesting to scientists within the field.

I have some questions and remarks, which I ask the authors to address before I recommend a publication in Nature Communications:

(1) A crucial point is to ensure that the luminescence measured between exciton and trion is an intrinsic property of the material. Can the authors give evidence that this emission is not related to defects or a result of the specific choice of the environment? In particular, hBN is known to obey a rich defect-related spectrum itself.

We thank the referee for the positive feedback. We have confirmed that no defect emission is observed in the hBN only area in our experimental conditions. Also, all reported spectral features have been repeatedly observed in multiple samples. For a more detailed answer, please see our reply to a similar question posed by the first referee (question 1).

(2) For the biexcitons, similar observations have been made recently for monolayer WS₂ (Nagler arXiv:1801.09255) as well as in pump probe experiments on WSe₂ (Steinhoff et al. arXiv:1801.04225). I think these two works should be mentioned and deserve a discussion, especially as their interpretation is different.

The paper authored by Nagler et al. follows the previous assignment by some of us on the biexciton peak (ref 7), which we now understand as the emission from a charged biexciton. We have notified the author about our new insight and look forward to their response.

The other one by Steinhoff et al. is an interesting paper combining both theoretical and experimental efforts on understanding the fine structure of biexcitons. Because the authors are focused on interpreting their pump-probe experiments, the spin-forbidden dark exciton is ignored, which is likely valid in their scenario. As a result, it is hard to compare our result to their conclusions, which are all about biexcitons involving spin-allowed transitions in a much higher density regime. In the updated manuscript, we have included it in our reference as an experimental work suggesting similar biexciton binding energies independent of the spin configuration, and it will be interesting to see how their theory can be extended to predict the bright-dark biexcitons in the future.

(3) The resonance the authors identified as a charged biexciton has recently been explained by the coupling of excitons to inter-valley plasmons (Tuan et al. PRX 7, 041040 (2017)). Can the authors rule out this alternative interpretation?

We have carefully reviewed the PRX paper authored by Tuan et al. The theoretical interpretation is about a relevant effect but not the same as our observed charged biexciton emission. Although both peaks are spectrally located in a similar energy range, the coupled plasmon-exciton model is aimed to explain the so-called optical side band ($X^{\prime-}$), which emerges and red-shifts with electron doping. Most importantly, the $X^{\prime-}$ emission persists even at a doping level up to 10^{13} cm^{-2} and becomes the dominant PL feature in that regime (fig 1c in the PRX paper and Wang et al. Nano Lett. 17 (2) 2017). In contrast, our observed charged biexciton is composed of a trion and an exciton, and therefore emits most strongly at low doping levels where both species exist (the doping density needs to be lower than $2 \times 10^{11} \text{ cm}^{-2}$). When the doping level is increased, the XX^- peak quickly vanishes, leaving a much weaker and persistent peak, which is potentially the plasmon-exciton hybridized feature as described in the PRX paper.

We have summarized the discussion above in the manuscript and included the reference.

(4) Why does the $X0$ transition broaden in case of p-doping and vanishes under n-doping?

The referee raises a very interesting point. [REDACTED] Experimentally, we observe the $X0$ emission becomes stronger as it gets broadened, w [REDACTED], until a very high doping level is reached. This is in contrast to the n doped regime where the exciton peak becomes monotonically weaker with doping. [REDACTED]

(5) The central argument for the dark exciton contribution to the biexciton is that under magnetic field the higher Zeeman splitted transition is stronger populated. The authors also claim that the charged

biexciton is formed by a trion and a bright exciton. According to their argument, I would expect a stronger low energy branch - is this indeed the case for the charged biexciton?

Compared to other main peaks, the charged biexciton has a broader linewidth and an asymmetric lineshape, the origin of which could potentially be associated with multiple biexciton species. As a result, we cannot draw any conclusion about the Zeeman splitting of the XX- at this stage.

Generally speaking, the charged biexciton is a much more complicated quasi-particle than the neutral biexciton. There are two different possible spin-valley configurations for the charged biexciton, assuming that there is no more than one electron/hole in any band: The first has two holes in opposite VB valleys, two electrons in the opposite lower CB valleys and one electron in one of the upper CB valleys. The second configuration has one electron in the upper CB instead of the lower CB.

Meanwhile, both configurations can be formed in different ways, as the extra charge could have been bound to either of the two exciton species involved in the formation process (see sketch below). For instance, the former configuration can be formed by a dark exciton and an intra-valley trion or, similarly, an inter-valley dark trion could bind to a bright exciton to form the same state. In addition, even more pathways can exist during the emission process. If we assume the observed emission to result from the bright exciton-hole recombination, the remaining particles can have various combinations, ranging from an inter/intra-valley trion to an exciton plus a free electron, leading to different total energies in the system.

We have abridged the above discussion into a new sentence in the main text without [REDACTED]

“In contrast to the negatively charged trions, no fine feature can be resolved in the $XX^{\wedge-}$ peak at this stage. If no more than one electron/hole resides in any band, there should be two spin-valley configurations for $XX^{\wedge-}$. Due to conduction band splitting, we expect the energetically favored configuration to be two holes in opposite valence band valleys, two electrons in opposite lower conduction band valleys and one electron in the higher conduction band....”

(6) In my opinion, it makes no sense to use biexcitons in TMDs as sources for entangled photons if the biexciton state is extended and not localized like in a quantum dot. Therefore, I find the statement related to Ref. 9 in the introduction a bit misplaced.

We have deleted the corresponding sentence, in line with the referees' suggestion.

Reviewer #3 (Remarks to the Author):

The manuscript by Ye and co-workers reports on the observation of neutral and charged biexcitons in monolayer WSe₂ encapsulated into high quality hBN layers. The interpretation is supported by continuous wave photoluminescence experiments performed at low temperature as a function of excitation power, out of plane magnetic field and electrostatic doping. This work is timely as similar observations have been made in WS₂ monolayers by the group of Tobias Korn (arXiv 1801.09255) and the broad community of researchers working on TMDC will certainly find interest in this work. I really appreciate the manuscript and how the authors provide additional explanations that raise open questions instead of leaving them under the carpet. Thus, I think these results will still be debated in the community as they seem to be in contradiction with the interpretation of Korn's group (arXiv 1801.09255) in WS₂ monolayer and with the pump probe experiments performed by Elaine Li's group (arXiv 1801.04225). I think that these two studies deserve to be mentioned in the text to nuance affirmative conclusions. Moreover, I have a few questions and suggestions before recommending publication in Nature Communications journal.

We are very grateful to the referee's encouraging comments. We now refer to the latest paper from Prof. Li's group in the current manuscript. (For detailed comparison, please see our reply to the 2nd question of referee #2) For the latest Korn group's paper, we think the contradiction is due to the fact they followed our previous erroneous assignment about the emission of the charged biexciton as a neutral biexciton. We have notified the authors about our new understanding on the biexciton binding energy.

[REDACTED]

- I think that Fig2b and 2c are not clear. For Fig2c, I suggest to put the biexciton at the good energy which is EX0+EXD-Ebinding. It will help the reader to understand why the amplitude ratio of XX is compared with the one of X0/XD on Fig2b.

We have changed the figure accordingly and hope that this representation can be understood more easily.

- In the supplementary information, the authors calculate a rough estimate of the exciton density needed for biexciton formation. This is very interesting and I understand that this is just a rough estimation, but I think that the values of the diffusion coefficient extracted from the linewidth are highly questionable. I suggest to use the very recent values measured in this publication (arXiv 1802.09201 (2018)).

We thank the referee for understanding that our calculation related to the exciton diffusion is a very rough estimation. After we use the large diffusion coefficient (205 cm²/s) reported by Cadiz et al. as the referee suggested, the threshold density for biexciton formation becomes significantly reduced, to a level comparable to the maximal bright exciton density we can achieve, if we continue to assume a 100% conversion efficiency. Of course in the real sample the conversion efficiency should be far below unity and the dark exciton density is always orders of magnitude higher than the bright exciton, merely due to the longer lifetime. Therefore, our main conclusion on the bright-dark biexciton composition is not affected. Meanwhile, we also notice a newly accepted PRL paper (arXiv 1804.09386 (2018)) reports that the measurable diffusion coefficient is highly dependent on the excitation power (varying from 0.3 to 30 cm²/s in their case) due to peak flattening effects caused by Auger recombination. We think more detailed study about the exciton diffusion can further improve our understanding on the biexciton formation process. The discussion above has been reflected in the updated manuscript and supplementary information as cited below.

“The bright-bright configuration is unlikely to occur given our experimental conditions: At the largest applied power density, we estimate an upper limit of $52 \text{ } \mu\text{m}^{-2}$ of bright excitons in the steady state, given the 2-ps lifetime of the bright exciton \cite{Robert2016} and assuming a 100% conversion efficiency from absorbed photons to bright excitons. This is even smaller than the dark exciton density at our lowest excitation level, as the dark exciton has a lifetime more than 50 times longer than the bright species, as reported by Robert et al. \cite{Robert2017} and confirmed in our sample. The low bright exciton density renders the bright-bright biexciton much less likely to form. (A rough diffusion model is set up in the supplementary information to elaborate the biexciton formation process) Therefore we conclude the observed biexciton is combined of a bright and a dark exciton (Fig 1 b).”

“In this section, we elaborate how diffusion contributes to the efficient biexciton formation. [...] We note this is a very rough estimation as the dark exciton diffusion coefficient has been reported to be as high as $205 \text{ cm}^2/\text{s}$ in the encapsulated sample at low temperature \cite{Cadiz2018}, which can substantially reduce the threshold density to $4 \text{ } \mu\text{m}^{-2}$. In addition, the Auger process also needs to be taken into consideration as the exciton density becomes large \cite{Kulig2018}. [...]”

- Can the authors propose an explanation of the superlinearity of the trions (exponents 1.2 in the SI).

The slight superlinearity in the power dependence of the trion as well as that of the exciton (exponent is also about 1.2 in the intrinsic regime) have been attributed to the partial filling of the defect states under weak excitation conditions (Ref 7). As the defects are gradually filled, the loss of excitons/trions to the defect states is reduced, thus giving rise to a superlinear power dependence in the emission intensity. We have summarized the discussion in the supplementary information.

- Minor: in page 4, can the authors add the value of the measured FWHM ?

The linewidth (FWHM) of the X0 exciton is $\sim 4.5 \text{ meV}$. High-resolution spectra give a linewidth of XD of smaller than 1 meV . We have added these numbers in the manuscript.

REVIEWERS' COMMENTS:

Reviewer #1 (Remarks to the Author):

The changes introduced by the authors make the manuscript now very clear and convincing. In my opinion it can be published in Nat. Commun. without further modification.

Reviewer #2 (Confidential Remarks to the Editor).

Reviewer #3 (Remarks to the Author):

The authors have replied to my questions and I recommend this manuscript for publication in Nature Communications journal.